# Lansoprazole Increases Inorganic Pyrophosphate in Patients with Pseudoxanthoma Elasticum: A Double-Blind, Randomized, Placebo-Controlled Crossover Trial

**DOI:** 10.3390/ijms24054899

**Published:** 2023-03-03

**Authors:** Belén Murcia Casas, Juan Luis Carrillo Linares, Isabel Baquero Aranda, José Rioja Villodres, Vicente Merino Bohórquez, Andrés González Jiménez, Miguel Ángel Rico Corral, Ricardo Bosch, Miguel Ángel Sánchez Chaparro, María García Fernández, Pedro Valdivielso

**Affiliations:** 1Internal Medicine Unit, Hospital Universitario Virgen de la Victoria, 29010 Málaga, Spain; 2Instituto de Investigación Biomédica de Málaga (IBIMA), 29010 Málaga, Spain; 3Ophtalmology Unit, Hospital Universitario Virgen de la Victoria, 29010 Málaga, Spain; 4Centro de Investigaciones Médico-Sanitarias (CIMES), Universidad de Málaga, 29071 Málaga, Spain; 5Pharmacy Unit, Hospital Universitario Virgen Macarena, 41009 Sevilla, Spain; 6Internal Medicine Unit, Hospital Universitario Virgen Macarena, 41009 Sevilla, Spain; 7Dermatology Unit, Hospital Universitario Virgen de la Victoria, 29010 Málaga, Spain; 8Department of Medicine and Dermatology, University of Málaga, 29016 Málaga, Spain; 9Department of Phisiology, Universidad de Málaga, 29016 Málaga, Spain

**Keywords:** pseudoxanthoma elasticum, inorganic pyrophosphate, lansoprazole, tissue-nonspecific alkaline phosphatase

## Abstract

Pseudoxanthoma elasticum (PXE) is characterized by low levels of inorganic pyrophosphate (PPi) and a high activity of tissue-nonspecific alkaline phosphatase (TNAP). Lansoprazole is a partial inhibitor of TNAP. The aim was to investigate whether lansoprazole increases plasma PPi levels in subjects with PXE. We conducted a 2 × 2 randomized, double-blind, placebo-controlled crossover trial in patients with PXE. Patients were allocated 30 mg/day of lansoprazole or a placebo in two sequences of 8 weeks. The primary outcome was the differences in plasma PPi levels between the placebo and lansoprazole phases. 29 patients were included in the study. There were eight drop-outs due to the pandemic lockdown after the first visit and one due to gastric intolerance, so twenty patients completed the trial. A generalized linear mixed model was used to evaluate the effect of lansoprazole. Overall, lansoprazole increased plasma PPi levels from 0.34 ± 0.10 µM to 0.41 ± 0.16 µM (*p* = 0.0302), with no statistically significant changes in TNAP activity. There were no important adverse events. 30 mg/day of lansoprazole was able to significantly increase plasma PPi in patients with PXE; despite this, the study should be replicated with a large number of participants in a multicenter trial, with a clinical end point as the primary outcome.

## 1. Introduction

Pseudoxanthoma elasticum (PXE, OMIM 264800, ORPHA 758) is a rare disease caused by biallelic variants of the ABCC6 gene, which encodes an ABC transporter primarily found in the hepatic and renal tissues [1]. As a consequence, less ATP is available to be transformed into inorganic pyrophosphate (PPi), which inhibits the formation of hydroxyapatite and the ectopic calcification of soft tissues. Thus, the disease involves the calcification of elastic fibers, inducing changes in the skin (from papules to cutis laxa), retina (angioid streaks progressing to neovascularization and blindness), and arteries (especially in the lower extremities, with intermittent claudication) [2].

In addition to the ABC proteins, PPi levels are regulated by two enzymes: tissue-nonspecific alkaline phosphatase (TNAP), which transforms PPi into two molecules of Pi, and the ectonucleotide pyrophosphatase/phosphodiesterase (ENPP1) that produces PPi from ATP. We and others have shown that TNAP activity is increased and PPi is reduced in PXE subjects compared with controls [3,4,5]. Local PPi levels also depend on the transmembrane protein progressive ankylosis protein homolog (ANKH), a protein which is an ATP transporter, and also on NT5E gene-encoding CD73, which hydrolyzes AMP to adenosine and inorganic phosphate (Pi) [6].

There is no specific treatment to stop progressive calcification in PXE. PPi supplementation has been tested in animal models through its addition to drinking water [7] and in humans by ingestion of gelatin capsules [8]. Furthermore, the inhibition of TNAP with SBI-425 [9] or the addition of recombinant ENPP1 [10] have been also tested. All of these have been shown to increase PPi levels and reduce mineralization. However, none of these therapies has been tested in human clinical trials.

There have been few clinical trials in PXE. One trial in PXE patients with oral magnesium supplementation for 2 years was not able to show benefit in terms of reducing calcification of the skin and there were no changes in retinal signs [11]. The treatment of etidronate for one year was compared with a placebo in a randomized trial, showing small but significant benefit in terms of reducing the progression of vascular calcification measured via CT but neither reducing the active deposition of Na18F in the femoral artery wall; no changes in the retina were noticed [12,13].

Lansoprazole is a proton-pump inhibitor licensed to treat diseases related to gastric acid output. Lansoprazole has been shown to partially inhibit the effects of TNAP [14] and PHOSPHO1 [15], two enzymes involved in mineralization. The aim of this study was to test in a clinical trial the effect of lansoprazole on plasma PPi levels in subjects with PXE.

## 2. Results

This is a double-blind, randomized, placebo-controlled crossover clinical trial. It was registered as ClinicalTrials.gov Identifier: NCT04660461. EudraCT Number: 2016-004021-16.

Patients with PXE were invited to participate through the web page of the Spanish Association of Persons Affected by PXE (www.pxe-espana.com (accessed on 20 December 2022)). Inclusion criteria were an age older than 18 years and diagnosis of PXE according to modified Plomp criteria [16], namely, at least two among three of the following: (1) retinal anomalies such as peau d’orange and/or angioid streaks; (2) skin anomalies such as yellow papules or plaques in the lateral side of the neck or in the flexures of the axillae, elbows, or knees, or fragmentation and/or calcification of elastic fibers on a skin biopsy; and (3) biallelic pathogenic variants at the ABCC6 gene. The exclusion criteria were the following: declined to sign the informed consent, vegetarian or extreme diets, pregnancy or intention to be pregnant during the trial, hypersensitivity to proton-pump inhibitors, drugs interfering with lansoprazole, and those who previously took a proton-pump inhibitor unless 15 days of clearance had elapsed if clinically allowed.

Patients entering the trial were randomized with 30 mg of lansoprazole daily or a placebo in the morning during a period of 8 weeks; after 2 weeks of washout, patients were allocated to a placebo or 30 mg of lansoprazole for another 8 weeks, as shown in Figure 1.

Plasma PPi levels were reported from 1.25 ± 0.25 µM in healthy controls [17]. In our laboratory, we showed that controls had 0.61 ± 0.18 µM and PXE patients had 0.35 ± 0.15 µM [3]. Based on these reports we planned to recruit at least twenty patients for this study.

The trial started in February 2020. Twenty-nine patients fulfilled the inclusion criteria and none of the exclusion criteria. Eight were lost to follow-up at visit two due to the pandemic COVID-19 lockdown, and one patient withdrew in the first week of the trial due to epigastralgia. In all, 20 patients completed the trial (Figure 1).Clinical characteristics of the patients are summarized in Table 1; the patients showed the classical signs of the disease in the skin (papules), eyes (drusae and angioid streaks), and vasculature (absence of pulses), as well as symptoms (visual loss and intermittent claudication). The patients’ age was 49 ± 10 years. Twelve (60%) were women. Hypertension, dyslipidemia, and renal lithiasis were common among participants. Peripheral arterial disease was more common than coronary and cerebrovascular disease. As a consequence, antihypertensives and lipid-lowering drugs were the most commonly used medicines in the study population (see Table 1). Laboratory data, comparing the placebo and lansoprazole phases, show no differences in values between the two periods of treatment (see Appendix A).

Table 2 shows data on specific values related to PXE. Interestingly, the level of PPi, the main endpoint of the trial, was significantly increased during the period of lansoprazole administration (F 15.07, *p* = 0.001), from 0.348 ± 0.100 µM to 0.411 ± 0.138 µM (+18%, *p* < 0.05) (Figure 2a). The response to lansoprazole, however, was heterogeneous, decreasing in three patients and increasing in the rest (Figure 2b and Figure 3). Comparing plasma PPi at baseline and at the end of the lansoprazole period, we observed an increase from 0.345 ± 0.09 µM to 0.41 ± 0.146 µM (+19%, *p* < 0.05). In contrast, the enzyme values related to PPi levels, such as the TNAP and ENPP1 activity and amount, did not change after lansoprazole treatment. Interestingly, PPi levels did not differ between visits one and three, indicating there was no carry-over effect after the washout period (Appendix A).

3.We observed six adverse events during the trial (Appendix A); in only one case did the patient drop out from the study due to the adverse event.

## 3. Discussion

Our trial shows that administration of lansoprazole 30 mg increased PPi levels in patients with PXE compared with the placebo in a short period of 8 weeks. Because PPi is recognized as a molecule directly involved in the pathogenesis of calcification in these patients, lansoprazole might be considered for new clinical trials with clinical outcomes.

Of note, PPi levels were similar at visits one and three (Figure 3, and Appendix A), indicating there was no carry-over effect between treatment phases. Furthermore, the levels of PPi measured in the trial in the placebo phase were also similar to those reported in previous analyses performed in our laboratory in PXE subjects [3]. In our trial, PPi levels increased globally by a mean of 18% (median 20.2% (2.7–31.5%)) after lansoprazole, compared with the placebo; however, the response was heterogeneous, increasing in 17 out of 20 patients tested (Figure 2b) and showing a range between −34% to +85% (Figure 3). One reason for this might be a lack of medication compliance; however, all patients returned less than 20% of the drug dispensed. Moreover, it is well known that the effects of drugs are highly variable: this variability is in part genetically mediated and proton-pump inhibitors are not an exception [18]; thus, we can expect some variation among participants.

The rationale for our trial relies on the fact that TNAP activity is higher and PPi is lower in PXE patients compared with non-PXE subjects [3,4], and that TNAP activity could be partially inhibited by lansoprazole [14]. Despite this, we could not show any decrease in TNAP activity in plasma samples of patients during the two phases of the study. Perhaps the method used to assess TNAP activity in our trial was not sensitive enough to detect significant changes. On the other hand, some papers show no inhibition of TNAP by lansoprazole [19,20], suggesting that the effect of the drug on TNAP is dose-dependent. In addition, we did not see changes in the ENPP1 amount or activity, which play a major role in the conversion of ATP to AMP + PPi. A potential role of lansoprazole on the activity of CD73 and ANKH protein is merely speculative, as we did not find any reference linking these proteins to the drug. Furthermore, a recent paper showed that omeprazole was not able to inhibit the expression of CD73 [21].

Another interesting point concerning lansoprazole and PXE is the effect of the drug on PHOSPHO1, a cytosolic phosphatase highly expressed in osteoblasts and essential for bone mineralization [22]. Lansoprazole was shown to weakly reduce TNAP (2.4%) as compared with PHOSPHO1 (52.8%) [23]. In another study, the inhibition of PHOSPHO1 suppressed vascular smooth muscle cell calcification by 41.1%, in comparison with the concomitant inhibition of PHOSPHO1 and TNAP, which reduced it by 20.8% [24]. Recently, lansoprazole (as well as other proton-pump inhibitors) was shown to inhibit PHOSPHO1 activity and matrix mineralization in vitro [19]. We did not perform studies in our trial to measure PHOSPHO1 activity, but it might be an additional beneficial effect of lansoprazole on PXE patients.

There were very few adverse side effects, with six in the placebo phase and three in the lansoprazole phase, mainly gastrointestinal intolerance, and in only one case causing the patient to drop out of the study (in the placebo phase). Putting this into perspective, we should take into account the long-term side effects of lansoprazole: on the one hand, it has been shown to increase the risk of osteoporotic fractures under long-term use and a high dose [25]. On the other hand, there is an increased risk of vascular calcification in patients on chronic hemodialysis due to hypomagnesemia [26,27] and the impairment of skeletal mineralization during limb development [15]. Hypomagnesemia has also been reported in 19% of subjects in studies performed on the general population [28] However, neither chronic effect seems to be relevant in the context of lansoprazole therapy for PXE patients.

In our study, lansoprazole elevated PPi more than the placebo by an average of 18%, partially compensating for the 30% deficit the patients with PXE had relative to the controls, but without normalizing levels [3]. However, the number of PXE patients within the normal range for controls (0.61 ± 0.18) increased from five (25%) in the placebo phase compared to eight (40%) in the lansoprazole phase. In the mouse model of PXE, therapy with INZ-701 (a recombinant ENPP1 enzyme) at the lowest dose of 2 mg/kg normalized PPi, reaching the level measured in wild-type mice [10]. In the experiment with PPi in water, plasma PPi levels were increased threefold 10 min after the mice drank it, but returned to normal at 30 min [7].

We do not know if the observed 18% increase might provide a clinical benefit to patients, such as delaying the progression of the disease. Only a long-term trial with clinical end points might answer this question, or a mid-term trial with the deposition of Na18F in the neck skin as the end point [29,30]. The major advantages of lansoprazole in comparison with therapeutic alternatives (PPi in drinking water, INZ-701, or etidronate) are its wide availability, low cost, and the low incidence of side effects associated with its long-term use.

Our trial has several limitations that should be taken into account: the low number of participants, the short-term follow-up (only 8 weeks of therapy), the fact we do not know how many days it took for PPi to be increased after lansoprazole was taken, the absence of clinical values as end points, as well the heterogeneous response in some patients.

In conclusion, our trial supports the establishment of a new clinical trial with lansoprazole in PXE patients, with clinical outcomes as the major end points. It might be considered as monotherapy or as an adjuvant with a second medication, such as oral PPi ingestion.

## 4. Materials and Methods

Capsules of lansoprazole 30 mg or a placebo were provided by the Pharmacy Service of Hospital Virgen de la Macarena, Sevilla, Spain. The random sequence was created using the software Random Integer Generator (www.ramdom.org (accessed on 1 January 2018)). A list was built, and patients were consecutively assigned to treatment. The washout period was considered sufficient (lansoprazole’s half-life is up to 4.2 h) [31] to avoid the carry-over effect typical of crossover studies [32].

Patients who consented to participate in the trial had 4 visits to the hospital, just before and after the two 8-week periods. At each visit, demographics, clinical and anthropometric data, and the presence of any adverse event were obtained. At the beginning of each 8-week period, a bottle containing 59 capsules of lansoprazole 30 mg or a placebo was given to each patient; the bottle was returned at the end of each period and the capsules were counted.

At the first visit, age, sex, date of the first PXE symptoms, skin, vascular and eye stage of the disease, and comorbidities (hypertension, diabetes, dyslipidemia, renal lithiasis, cardiovascular or cerebrovascular disease, intermittent claudication, and any concomitant therapy) were recorded. Body weight, stature, body mass index, blood pressure, and the ankle–brachial index were registered. At each visit, the physical examination, medication, adverse events, and returned pills were recorded.

2.At each visit, after an overnight fast, blood samples were drawn and collected into serum, K2-EDTA, citrate, and citrate-theophylline-adenosine-dipyridamole (CTAD) vacuum tubes (BD Vacutainer; Plymouth, UK). Blood samples were kept on ice and then the K2-EDTA tubes were centrifuged for 15 min at 1750× *g* at 4 °C. Serum (off the clot) and plasma aliquots were obtained on ice and stored at −70 °C until assayed. CTAD blood tubes were centrifuged for 15 min at 800× *g* at 4 °C. Plasma was then transferred into separating Vivaspin 6 tubes 300,000 Molecular Weight Cut off (Sartorius, ref: VS0652; Stonehouse, UK) and filter-centrifuged at 3100× *g* for 35 min at 4 °C. Filtered plasma samples were stored at −70 °C until further processing. The ratio of charged CTAD plasma in Vivaspin 6 tubes to the eluted volume was considered in the final calculations.

PPi, TNAP and ENPP1 activities, CXCL4, Osteocalcin, and ENPP1 concentrations were measured as described previously [3].

Hematological and basic biochemistry variables were collected through a clinical autoanalyzer and the Siemens Dimension Vista System Flex analysis system at the clinical laboratory of the Virgen de la Victoria University Hospital in Malaga, Spain.

3.Statistical analyses: These were performed by a statistician not on the research team. The statistical package used was R 3.5.2. Data are shown as mean ± SD, median (IQR) or n (%). A generalized linear mixed-model approach was used to estimate differences between periods of lansoprazole and placebo while accounting for within-subject correlations arising from the crossover design. Study data were collected and managed using REDCap electronic data capture tools hosted at Hospital Regional Universitario de Málaga, Spain [33]. GraphPad Prism 9.4.1 was used to create graphs.

## Figures and Tables

**Figure 1 ijms-24-04899-f001:**
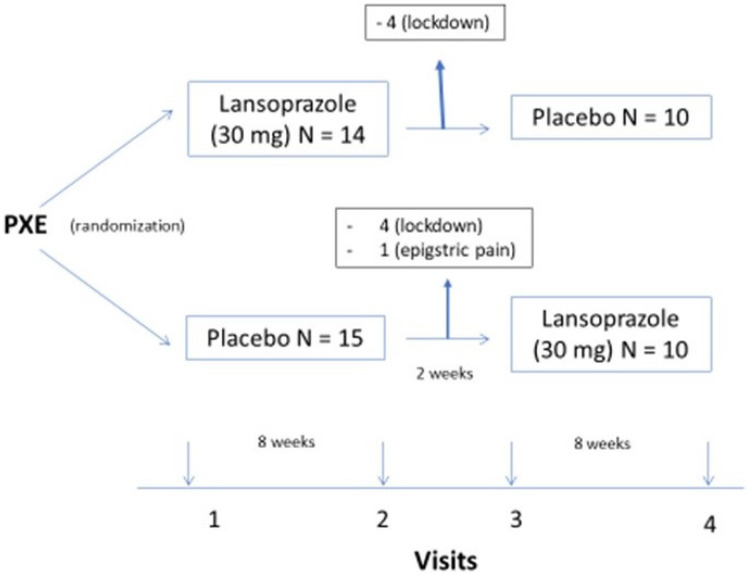
legend: design of the trial.

**Figure 2 ijms-24-04899-f002:**
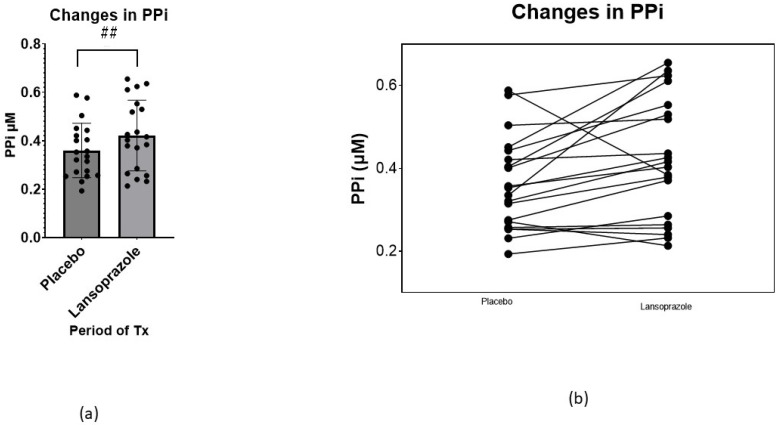
(**a**) Changes in plasma PPi comparing absolute changes at the end of the placebo phase and at the end of the lansoprazole phase (mean ± SD) (*p* < 0.05). (**b**) Individual absolute changes in PPi, comparing the end of the placebo phase and the end of the lansoprazole phase. ## = *p* < 0.05.

**Figure 3 ijms-24-04899-f003:**
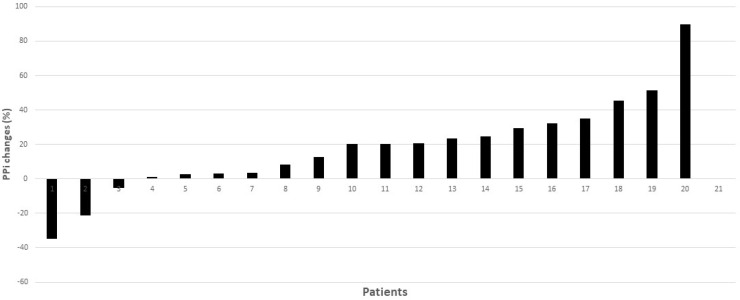
Individual percentage changes in PPi, comparing the end of the placebo phase and the end of the lansoprazole phase. 20.2% (2.7–31.5%).

**Table 1 ijms-24-04899-t001:** Clinical characteristics of patients.

Age (Years-Old)	49.4 ± 10.47
Women	12 (60)
Skin	
Papules	19 (95)
Plaques	17 (85)
Cutis laxa	10 (50)
Eye	
Angioid streaks	20 (100)
Peau d’orange	7 (35)
Drusae	2 (10)
Optic atrophy	5 (20)
Neovascular membranes	6 (30)
Artery	
Intermittent claudication	4 (20)
ABI < 0.9	2 (10)
Coronary heart disease	3 (15)
Stroke	2 (10)
Other symptoms and vascular risk	
Renal lithiasis	7 (35)
Smoking	3 (17)
Diabetes	1 (5)
Hypertension	8 (41)
Hypercholesterolemia	10 (50)
Concomitant medication	14 (70)

Data are showed as mean ± SD or N (%). Abbreviation: ABI, ankle–brachial index.

**Table 2 ijms-24-04899-t002:** Laboratory parameters in the final visits according to the phase of therapy.

	Placebo	Lansoprazole	*p*-Value
PTH	50.68 ± 20.32	43.97 ± 22.51	0.1892
PPi (µM)	0.34 ± 0.10	0.41 ± 0.16	0.0302
ENPP1 (activity) µU/L	5.58 ± 0.8	5.76 ± 1.71	0.0886
ENPP1 (amount) (ng/mL)	19.07 ± 5.23	19.25 ± 3.76	0.8065
Alkaline phosphatase (U/L)	70.10 ± 24.58	68.11 ± 19.09	0.1196
TNAP (U/L)	87.75 ± 27.69	88.65 ± 25.54	0.3322
CXCL4 (µg/mL)	1.90 ± 1.53	1.79 ± 1.49	0.7815
Osteocalcin (ng/mL)	14.24 ± 5.37	12.59 ± 4.09	0.0798

Abbreviations: PTH, parathyroid hormone; PPi, plasma inorganic pyrophosphate; ENPP1, ectonucleotide pyrophosphatase/phosphodiesterase; TNAP, tissue-nonspecific alkaline phosphatase; CXCL4, platelet factor 4.

## Data Availability

Data will be provided upon reasonable request.

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
