# Peer review of "Lansoprazole Increases Inorganic Pyrophosphate in Patients with Pseudoxanthoma Elasticum: A Double-Blind, Randomized, Placebo-Controlled Crossover Trial"

_ijms, 2023, doi:10.3390/ijms24054899_

Round 1

Reviewer 1 Report

This is a double blinded, randomized, placebo controlled clinical trial with crossover design that investigates the effect of proton pump inhibitor Lansoprazole on levels of pyrophosphate (PPi) and its regulatory mediators in patients with Pseudoxanthoma elasticum, where PPi levels are known to be reduced and thus increases calcification. The main finding of the study is that PPi levels were significantly increased after Lansoprazole treatment. The trial has a strong cross-over design, and the results are interesting and could stimulate important further studies, evaluating long term effects of treatment. However, I find the manuscript lacking important information in several sections in its current form, making it difficult to fully interpret study results. My specific comments are:   

Major comments:

1. The aim in the manuscript differs from the aim in the clinical trials registration. From verifying the changes in plasma PPi, and the main molecules that regulate it (NPP1-3, TNAP) in the clinical trials registration to only changes in PPi in the manuscript.

 2. The authors don’t specify in methods at what time points PPi, NPP1-3 and TNAP was measured in blood samples.

3. The authors don’t show data from measurements at visit 1 and 3 stating that there was no difference in concentration between the time points (washout period) while the main comparison was done between weeks 2 and 4. I would recommend adding the full data as supplemental material so that readers are able to see the full measures made throughout the trial. 

4. The authors don’t specify how the randomization was performed. Method used to generate the random allocation sequence, type of randomization, who created the randomization method and who randomized the patients should be included for example.

5. Methods don’t include information on blood samples reported in supplementary figure 2 

6. Not all parameters presented in table 2 are mentioned in the methods section.

7. Statistical methods need to be further specified to include full details. There are methods missing for statistical testing for reported p values in table 2, and a description of what groups were compared here. Specify versions of statistical software used. If I’m not mistaken it looks like Graphpad prism was used to compute the graphs of figure 1. This is not mentioned. The R package used for the mixed linear model could also be included.

8. Two figures labeled as nr 3. In the figure 3 of study design, include how many patients was allocated in each arm in the trial. Include the 9 patients that were excluded.

9. Figure 1 a) b) and figure 3 below it illustrates the same data. Figure 3 shows the same information as fig 1b in proportions and can be removed.

10. The comparisons made for the analysis in Table 2 and Figure 1 seems to be different and is not clearly explained. Table 2 seems to compare groups means between treatment groups at the end of the trial while in fig 1 uses repeated measures of the same individual. This needs to be clarified. Why was only PPi evaluated in your mixed model and not for the other regulators? I would expect more power from this comparison instead of comparing group means like in table 2. 

11. There are some inconsistencies in the numbers reported at end of Lansoprazole in row 98 and 102 with different SD for what should be the same data?

12. The discussion can be expanded regarding how PPi was upregulated in absence of changes of its regulatory mediators. The authors mention that the method to measure TNAP might not have been sensitive enough, but this might also have been an issue of low power. Is there any other mechanisms of which Lansoprazole could have increased PPi?

Minor comments:

13. Table 1: Include what numbers represent for each variable (counts/SD). Describe what concomitant therapy means.

14. Row 107: Analytical misspelled.

15.  Units in table 1 missing

Reviewer 2 Report

A brief summary This article has succeeded to prove the hypothesis that lansoprazole compared with placebo in a cross-over manner increases plasma pyrophosphate.

General comments
This article soundly tests with suitable methods the hypothesis that lansoprazole increases plasma pyrophosphate compared with placebo. As a cross-over study it is feasible that the patients serve as their own controls. The manuscript is mostly clearly written, answers a relevant question and is presented in a well-structured manner.

Specific comments 

  • Within the manuscript there is slight confusion because study phases are called study sequences in other parts (for example supplementary table 3). Also the word periods is used for the same in line 89. Please synchronize terminology for easier reading.
  • Line 28-29: The abbreviation for pyrophosphate (PPi) is switched to cPPi in many places within the manuscript. Correct.
  • Conclusions of the abstract: Lines 39-41: As the results of the abstract do not give the reference range for plasma PPi, either in the results or in the conclusions it should be mentioned that lansoprazole was not able to normalize the concentration. I also find it a too strong expression to say that lansoprazole should enter larger clinical studies considering other potential treatments under investigation. Tune down the message slightly.
  • Line 48: Instead of two pathogenic variants, biallelic pathogenic variants would give more information.
  • Line 51: sentence “…formation of hydroxyapatite and calcification of tissues.” I suggest to clarify by adding two words “…formation of hydroxyapatite and ectopic calcification of soft tissues.”
  • Line 53: “…vessel walls…” wouldn’t it be more correct to mention arteries instead of vessels? Is there proof that venous structures are also calcified?
  • Lines 55-59: I would very much like to see the ANKH gene/protein mentioned here with a profer reference. Also CD73 protein could be important especially because the increased TNAP activity is mentioned. These both effects can be seen for example in the Figure 3 of the publication Jansen et al. Arterioscler Thromb Vasc Biol. 2014;34:1985-1989.
  • Line 60: “There is no specific treatment for PXE.” How about adding the idea of …to stop the progressive calcification in PXE.””
  • Lines 60-63: The sentence and referencing thereby is partly erroneous. Gelatin capsules have never been given to animals but instead to humans. Also SBI-425 and ENPP1 are not therapies of supplementation but instead enzyme inhibition and enzyme replacement therapies, respectively.
  • Line 66: nonsignificant benefit. This cannot be used. If there is no significant change, there is no benefit. “Was not able to show benefit” would be more suitable.
  • Line 70: The Na18F-PET scanning did not show any change in this study. Please correct the referencing.
  • Line 73-74: If the information is available, I would like to see percenteges about how much inhibition does the partial inhibition mean for both TNAP and PHOSPHO1.
  • Line 83: Although the word symptoms is much used to refer to both subjective symptoms and objective findings/signs of a disease, I suggest to make the difference clear here. Which is it symptoms or signs?
  • Lines 83-85: Data about patient age and gender are given both in the text and Table 1, but data about hypertension and dyslipidemia in neither.
  • Line 87: Antihypertensives instead of hypotensives?
  •  Line 87: lipid-lowering drugs were the most commonly used medicines in the study population?
  • Line 88: Analytical. Would laboratory be more fitting?
  • Line 88: the sentence is about medications but referring to supplemental Table1 is erroneous since the Table 1 is about physical examination.
  • Table 1: Would Concomitant therapy be better said with Concomitant medications or Concomitant medical therapy or does this refer to other therapies than just medicines.
  • Line 104-105: Would it be more clear if in the parentheses in the end one would mention Figure 3 where the phases are shown.
  • Table 2 title: The title needs revising. I don’t think that placebo should be named as a drug. I suggest to use study phases. The same goes for the title of supplemental Table 2.
  • Table 2: Similarly as in the text I suggest to use PPi instead of cPPi.
  • Line 118: Correct Figure number from 3 to 2.
  • Line 129: again I suggest to add Figure 3 in parentheses when study phases are mentioned.
  • Lines 137-140: The sentence is rather long. I suggest to discard the part about statins (with reference) and to concentrate on the pharmacogenetics of lansoprazole. This would make the sentence more readable.
  • Lines 144-145: I suggest to use study phases in this sentence instead of referring to patients treated with lansoprazole and those with placebo, because all patients received both in different study phases.
  • Lines 166-167: the risk of hypomagnesemia in not only patients with chronic hemodialysis by long-term use of proton pump inhibitors should be addressed. For example this reference could be useful: Mechanisms of proton pump inhibitor-induced hypomagnesemia. Gommers LMM, Hoenderop JGJ, de Baaij JHF. Acta Physiol (Oxf). 2022 Aug;235(4):e13846. doi: 10.1111/apha.13846. Epub 2022 Jun 14.
  • Line 173: her to make terminology different between the increase/elevation of PPi by lansoprazole and normalization of PPi by INZ-701, I would use the word normalized rather that elevated. The non-normalization of plasma PPi by lansoprazole should receive more attention in the article elsewhere also
  • Limitations, line 184-186. My suggestion is to add a limitation that this study was not able to tell how soon the PPi-increasing effect starts, because now we know what is the effect at the 8-week time point.
  • Lines 188-189: I would be more careful about the future. As science goes, it would first be more important that the results be reproduced before rushing to a clinical trial. It could turn out that since the normalization of plasma PPi cannot be achieved by lansoprazole, it could turn out to be used as an adjuvant therapy?
  • Lines 192-194: In the article there is no mentioning about complying with the Declaration of Helsinki.
  • Line 206: check spelling of the internet address
  • Line 241: It would improve the manuscript a lot if the achieved plasma PPi elevation would be compared to this reference range more in the results and the discussion. For example what was the percentage below the lower limit of normal during the placebo phase and during the lansoprazole phase.
  • Line 242: This sentence about the favorable response is unintelligible. 
  • General comment: I suggest to check the use of words outcome versus endpoint  in the text.
  • Supplemental Table 3: the numbers and texts in parentheses should be alike in the columns.

Round 2

Reviewer 1 Report

The authors have addressed the points raised.

Author Response

There is no new suggestion by the reviewer